# Photochemical Reduction of Silver Nanoparticles on Diatoms

**DOI:** 10.3390/md21030185

**Published:** 2023-03-17

**Authors:** Adrián León-Valencia, Sarah Briceño, Carlos Reinoso, Karla Vizuete, Alexis Debut, Manuel Caetano, Gema González

**Affiliations:** 1School of Physical Sciences and Nanotechnology, Yachay Tech University, Urcuquí 100119, Ecuador; 2Centro de Nanociencia y Nanotecnología, Universidad de las Fuerzas Armadas ESPE, Sangolqui, Quito 171103, Ecuador; 3School of Chemical Sciences and Engineering, Yachay Tech University, Urcuquí 100119, Ecuador

**Keywords:** diatom, nanoparticles, SERS, fluorescence, biosensor

## Abstract

In this work, the photochemical reduction method was used at 440 or 540 nm excitation wavelengths to optimize the deposition of silver nanoparticles on the diatom surface as a potential DNA biosensor. The as-synthesized nanocomposites were characterized by ultraviolet-visible spectroscopy (UV-Vis), Fourier transforms infrared spectroscopy (FTIR), X-ray photoelectron spectroscopy (XPS), scanning transmission electron microscopy (STEM), fluorescence microscopy, and Raman spectroscopy. Our results revealed a 5.5-fold enhancement in the fluorescence response of the nanocomposite irradiated at 440 nm with DNA. The enhanced sensitivity comes from the optical coupling of the guided-mode resonance of the diatoms and the localized surface plasmon of the silver nanoparticles interacting with the DNA. The advantage of this work involves the use of a low-cost green method to optimize the deposition of plasmonic nanoparticles on diatoms as an alternative fabrication method for fluorescent biosensors.

## 1. Introduction

Biological nanomaterials have gained importance in biomedicine due to their large variety of bioresources, synthesis methods, versatility, and the possibility to further enhance their features through material functionalization [1,2]. They have become the focus of the next generation of biomedical tools, such as drug delivery, target-specific nanomaterials, biosensors, and bioimaging materials [3]. In matters of biosensing and bioimaging materials, diatoms are great options due to their particular structures, characteristic shapes, replicability, luminescent properties, and biocompatibility [4].

Naturally occurring biosensors doped with plasmonic nanoparticles are increasingly used to detect physiologically important analytes in real biological samples (i.e., blood, plasma, urine, and saliva) [5,6]. Biosensors are materials that detect single molecular species by combining the molecular recognition properties of biological macromolecules with signal transduction mechanisms that couple ligand-binding to readily detectable physical changes. The choice of the substrates for the analytical applications depends on the transduction methods, i.e., fluorescence or chemiluminescence, and the application of the measurements. In recent years, fluorescent biosensors based on biomaterials have been reported using organic dyes, inorganic semiconductors [7], carbon quantum dots [8], and diatoms as fluorescence biosensors [9].

Diatoms are low-cost and biocompatible 3D materials that show interesting and remarkable optical properties [10,11]. Diatoms have been applied in two main monitoring techniques in biosensing applications, i.e., fluorescence and surface-enhanced Raman scattering (SERS) [12]. Fluorescence immunoassays based on diatoms are developed by cross-linking antibodies on the surfaces of diatoms. A significant increase in the photoluminescence intensity was observed upon biomolecular interactions with the analyte, indicating the biosensing response of the diatoms [13]. It has been reported that incident light on a diatom frustule generates enhanced electromagnetic fields on or near the surface of the diatom structure [14]. This results in increased excitation of the fluorophore and emission of more photons. Additionally, a fluorophore on the surface of a frustule experiences an increase in the density of optical states, which results in enhanced emission due to the Purcell effect [15,16]. This dual-modal optical enhancement results in boosted fluorescence signals for the easier detection of analytes [17]. Due to their unique structural properties, diatoms are capable of enhancing the localized surface plasmon resonance (LSPR), leading to near-field optical amplification of the signal [17] due to the photonic, hierarchical, micro-scale periodicity of the diatom frustule pores that allow for the signal enhancement of the electromagnetic field induced by LSPR ‘hot spots’ with large SERS enhancement factors [14]. As a consequence, diatom biosilica has found wide applications in the fields of biosensing [18], diagnostics [19], and therapeutics.

Several methods have been reported to optimize the properties of diatoms, as their surfaces can be functionalized and decorated with different materials depending on the desired characteristic to be improved [2,20]. These properties can be improved by the functionalization and decoration with plasmonic nanoparticles as potential surfaced enhanced Raman scattering (SERS) biosensors [21,22], photoluminescent immunosensor [18], and multifunctional drug delivery carriers [4]. Taking advantage of the intrinsic luminescence of diatom frustules and the plasmonic resonances at visible wavelengths associated with the intrinsic surface plasmon frequencies of metallic nanoparticles [23], these nanocomposites could be potential materials for biosensors [14,19]. It has been reported that natural diatom biosilica combined with metallic nanoparticles could be used in sensing, diagnostics, and therapeutics, taking advantage of the fluoresce and the SERS effect [24].

In this work, a simple low-cost photoreduction method to optimize the deposition of silver nanoparticles on diatoms was employed and a preliminary study as a potential DNA fluorescence-detecting biosensor was carried out. The novelty relies on controlling the silver nanoparticle growth and deposition on the diatom surface by irradiating the samples with 440 or 540 nm excitation wavelengths, respectively. The fluorescence properties and plasmonic resonance signals of silver nanoparticles on diatoms were enhanced, demonstrating potential applications as DNA fluorescence-detecting biosensors.

## 2. Results

### 2.1. Ultraviolet-Visible Spectroscopy (UV-Vis)

After the photochemical reduction process with 540 nm or 440 nm wavelengths, the sample solutions changed from colorless to purple for the samples irradiated with 440 nm (Ag Blue) and yellow for the sample irradiated at 540 nm (Ag Green). Figure 1 shows the absorption spectra of the silver nanoparticles on diatoms after the photochemical process. In these spectra, the absorption peak in the UV range from 200 nm to 300 nm was due to diatom frustules in both samples [25]. The photoreduction of the silver nanoparticles over the diatoms is evidenced by the surface plasmon resonance (SPR) peak observed at 424 nm from the excitation of free electrons of the spherical-shaped silver nanoparticles [4]. To complement the structural analysis of the diatom, the surface area and pore volume were measured using the Brunauer–Emmett–Teller technique, obtaining 102.30 m^2^/g and 0.22 cm^3^/g, respectively. The diatom surface provides a suitable environment for the photochemical reduction of the silver nanoparticles on the diatom surface.

### 2.2. X-ray Photoelectron Spectroscopy (XPS)

X-ray photoelectron spectroscopy was employed to measure the silver nanoparticle binding energies and atomic concentrations over the diatoms after the photochemical process. XPS measurements were carried out in the range of 0–1200 eV; the results are presented in Figure 2 and Table 1. The most intense peaks in the spectra correspond to O1s, Si2p, C1s, and Ag3d elements. Figure 2d shows the high-resolution scan of Ag3d, confirming the growth of silver nanoparticles over the diatom surface. In these spectra, we observe the highest percentage of AgO 86% in the diatom irradiated at 440 nm compared with the diatom irradiated at 540 nm at 78%. In Table 1, the atomic concentration of Ag3d is 0.8 % higher for the sample irradiated at 440 nm. The decrease in the intensity of the O1s peak at 532 eV for the samples after the photoreduction treatment suggests the interaction between the silver nanoparticles and the diatom surface. The results of the deconvolution analysis of the oxygen core level reveal a greater presence of oxygen–carbon bonding compared to oxygen–metal bonding in the Ag blue diatom due to the prevalence of functional groups. This is further substantiated by the higher atomic concentration of C1s, as depicted in Figure 2a,b. Based on the results obtained with XPS, we conclude that we obtained the highest concentration of silver nanoparticles on the diatom surface for the sample irradiated at 440 nm.

### 2.3. Scanning and Transmission Electron Microscopy (SEM and TEM)

Figure 3a shows the diatomite skeletal structure with a cylindrical and porous frustule with well-defined dimensions and an average of 25 μm in length, 10 μm in diameter, and a pore size of 0.9 ± 0.2 μm. In this figure, the effectiveness of the photoreduction process is revealed by the presence of silver nanoparticles on the surface of the diatoms. In Figure 3b, the nanoparticles irradiated with the 540 excitation wavelength show agglomerations with 10 ± 3 nm. Meanwhile, in Figure 3c, the diatoms irradiated at 440 nm excitation wavelengths show a high concentration of nanoparticles with 13 ± 5 nm and 9 ± 3 nm uniformly distributed over the diatom surface.

### 2.4. Fourier Transform Infrared Spectroscopy (FTIR)

In Figure 4, the most prominent peak of the diatom appears at 1003 cm${}^{-1}$ related to the Si–O–Si stretching, at 791 cm${}^{-1}$ to the Si-OH silanol stretching, and 444 cm${}^{-1}$ to the siloxane bending [26]. These spectra are characteristic of the siliceous structure of the diatom frustule and show that the photoreduction process does not alter the diatom structure. After the photoreduction process, we added a drop of DNA to the nanocomposites and characterized them using FTIR spectroscopy. In Figure 4 and Table 2, we identify the main peaks related to the interaction between the DNA and the nanocomposites through the N-H and C-N bindings at 2974 and 1044 cm${}^{-1}$ [27].

### 2.5. Raman Spectroscopy

Raman spectra of the diatoms are shown in Figure 5. Diatoms with silver nanoparticles (after photoreduction: Ag blue and Ag green) and DNA spectra show flat lines(Figure 5a–c,f, respectively). A drop of DNA was added to the nanocomposites, which resulted in an enhancement of the Raman intensity for the sample irradiated at 440 nm (Figure 5e), with 5.5 times higher detection sensitivity for DNA (Figure 5d). In these spectra, we identify the main peaks related to the DNA interacting with the nanocomposites through the N-H, C-N, and C-H bonds reported in Table 3, which are in good agreement with the results obtained with FTIR (Figure 4 and Table 2). Our findings are consistent with previous studies that have reported on applying Ag NPs on Pinnularia diatoms for SERS sensing using Rhodamine at 532 nm, resulting in a 4–6× improvement in sensitivity [28]. Additionally, gold-coated Aulacoseira diatoms were effective SERS substrates, exhibiting a significant enhancement of the spontaneous Raman scattering of p-Mercaptoaniline by a factor of 105 [29].

### 2.6. Optical and Fluorescence Microscopy

The fluorescence enhancement of the diatoms irradiated at 440 nm in Figure 6e is explained when the incident wavelength leads to the excitation of the surface plasmon, coherent electronic motion, and the d electrons [4]. When plasmonic nanoparticles are excited with molecular vibration results in an increase in the electrostatic fields surrounding the metallic nanoparticles on the diatom surface, this behavior is consistent with the observed surface plasmon band in Figure 1 in the visible region, due to the surface plasmon oscillation of free electrons and the high concentration of nanoparticles on the diatom, in good agreement with the XPS results in Figure 2 and the SEM images in Figure 3.

## 3. Discussion

In the first part, we evaluated the silver nanoparticle’s photoreduction over the diatom’s surface using ultraviolet-visible spectroscopy (UV-Vis) by the surface plasmon resonance peak at 424 nm [4]. With X-ray photoelectron spectroscopy, we found the highest percentage of AgO 86% in the diatom irradiated at 440 nm compared with the diatom irradiated at 540 nm with 78%, in agreement with the highest concentration of nanoparticles evidenced by scanning electron microscopy. Our results reveal that silver nanoparticles were deposited on the diatom surface, likely via an electrostatic interaction by the photoreduction process at 440 or 540 nm excitation wavelengths. Changing the irradiation wavelength using the photoreduction method allows the control of the nanoparticle size and distribution on the diatom surface. The growth of the nanoparticles could be explained when the nanoparticles are in resonance with the excitation wavelength forming rapid nucleation of Ag NPs on the diatom surface; this process is known as plasmon-assisted growth [30]. In this process, the $\lambda_{irr}$ dependence control the size and anisotropy of the nanoparticles. As shown in Figure 3d, larger sizes, and a wider distribution are obtained for the samples irradiated at 540 nm wavelength. Meanwhile, the 440 nm excitation wavelength promotes a gradual growth of the nanoparticles along the diatom surface with a uniform narrow distribution [31].

In the second part, we added a drop of DNA to the nanocomposites. We characterized them using FTIR and Raman spectroscopy, identifying the main peaks related to the interaction between the DNA and the nanocomposites through the N-H and C-N binding [32]. Finally, our results reveal that the highest concentration of Ag NPs on the diatom surface was obtained for the sample irradiated at 440 nm (Ag Blue), causing a surface-enhanced resonance Raman effect and the highest fluorescence response. The enhanced intensity observed comes from the optical coupling of the diatoms’ guided-mode resonance, the silver nanoparticles’ localized surface plasmon, and the DNA’s coupling on the nanocomposite surface. It has been reported [33] that the electromagnetic enhancement observed originates from two contributions; the local field enhancement and the radiation enhancement. The plasmon energy from the silver nanoparticles on the diatom forces the Raman process to occur in the DNA, the energy is transferred back into the plasmon, and the scattered radiation is detected. Another explanation could be related to the Hot spots formed by the interaction of the nanoparticles on the diatom surface. If the DNA interacts with the hot spot, the electromagnetic field increases, providing the surface-enhanced Raman scattering effect [34].

Previous studies [14] have demonstrated that the wavelengths of the localized surface plasmon (LSP) resonances at the metallic surfaces are determined by the overall geometries and the aggregation states of the Ag NPs. The concentration of Ag NPs induces plasmonic extinction at longer wavelengths when individual nanoparticles are in a close-packed assembly and coupled with each other. The frequency and intensity of the plasmon oscillation depend on the degree of agglomeration as well as orientation with respect to the polarization direction of the excitation light. In our case, the enhancement of the fluorescence response of the nanocomposite irradiated at 440 nm, was 5.5 times higher with DNA. This enhancement is due to the multiple resonances of similar aggregation states that have close resonant frequencies of the Ag NPs irradiated at 440 nm. This work demonstrates a way to design a surface-enhanced Raman spectroscopy (SERS) signal while simultaneously increasing the fluorescence signal through a combination of diatoms and silver nanoparticles. This behavior is particularly desirable for diagnosis and biosensing applications.

## 4. Materials and Methods

### 4.1. Chemicals

Diatoms were collected from the Guayllabamba inter-mountain basin, Ecuador, from the *planktonic* species and genus *Aulacoseira*. Hydrogen peroxide ($H_{2}O_{2}$), sulfuric acid ($H_{2}SO_{4}$, and trisodium citrate ($Na_{3}C_{6}H_{5}O_{7}$) were purchased from Loba Chemie. Silver nitrate ($AgNO_{3}$) and sodium borohydride ($NaBH_{4}$) were obtained from Sigma-Aldrich. The chemicals were not further modified and were used as received.

### 4.2. Diatom Extraction

Diatoms were separated from the rock using a scalpel, extracting the white layers to obtain a white powder (Figure 7a). Since diatoms probably contain significant residues from the rock, a purification method was performed (Figure 7b). Diatoms were treated with a piranha treatment with 80 mL of $H_{2}SO_{4}$ (1M) and 20 mL of $H_{2}O_{2}$, to clean the diatoms from the rock and eliminate all the organic compounds and pollutants. For this process, 7 g of diatoms were used. The piranha treatment was performed using a relation 4:1 of $H_{2}SO_{4}$ and $H_{2}O_{2}$, and the diatoms were immersed slowly in the solution for 30 min at 60 ${}^{\circ }$C with stirring. After the time-lapse, the solution was washed several times with water and, at the same time, tested with 0.5 M of NaOH solution. A white precipitate was obtained to indicate the presence of clean diatoms. The white powder sample was dry in the oven at 40 ${}^{\circ }$C–60 ${}^{\circ }$C for three days [20].

### 4.3. Photoreduction of Silver Nanoparticles on Diatoms

The photoreduction process was performed using irradiation of light at 440 or 540 nm excitation wavelength, respectively (Figure 8), following the design of Saade et al. [35] with further modifications previously reported [36,37] to provide the power of wavelength to the chambers using LEDs of 1 W each. Eight LEDs were placed in a series configuration with a current of 0.7 A and 5 V, inside a PVC tube of 12 cm in height, 11 cm in diameter, and 2 mm in thickness. To decorate the samples, 50 mg of diatoms were dispersed in 50 mL of $H_{2}O$. Then, 0.9 mL of sodium borohydride was added to the solution. Afterward, and with constant stirring, 75 μL of silver nitrate (0.1%) was injected each 5 min for 1 h. Furthermore, the samples were kept at constant stirring and exposed to the wavelength source inside the chambers for 4 h. The same process was repeated for both irradiation wavelengths (440 or 540 nm).

### 4.4. DNA Purification and Functionalization

A bacteria culture was performed using Gram-negative *E to obtain the DNA. coli* ATCC 25922, for 24 h at 37 ${}^{\circ }$C and constant shaking. Afterward, the purification of the DNA was performed using a Thermo Scientific GeneJET Genomic DNA Purification Kit. Bacteria cells were harvested in 1.5 mL and centrifuged for 10 min. Then, the pellet was suspended in 180 μL of their digestion solution, followed by adding 20 μL of proteinase K. After shaking thoroughly, the tube was incubated at 56 ${}^{\circ }$C in a shaking water bath for 30 min, followed by the addition of 20 μL of RNA solution mix and incubated for 10 min, and 200 μL of lysis solution was added and vortexed for 15 s. Then, vortexing mixed 400 μL of 50% ethanol. The sample was transferred to a DNA purification column, and the residue was centrifuged for 1 min at 6000 rpm. The sample was washed with 500 μL of the buffer solution, centrifuged for 3 min at 8000 rpm, and repeated three times before eluting the genomic DNA with the purification column. The DNA sample was incubated for 2 min at room temperature and centrifuged for 1 min at 8000 rpm. The purified DNA was stored at −20 ${}^{\circ }$C for later use. Finally, the DNA was attached to the decorated diatom by the dripping method with 1 μL of DNA on 0.1 g of decorated diatom at room temperature.

### 4.5. Characterization

The surface area was measured using the Brunauer–Emmett–Teller technique in an ASAP 2100 from Micromeritics. Infrared analysis was performed using an Agilent Technologies spectrometer Cary 360 with a diamond attenuated total reflectance (ATR) accessory and resolution of 4 cm${}^{-1}$. UV-Vis spectra were acquired with a LAMBDA 1050 UV/Vis/NIR, with Accessory Praying Mantis. Scanning Transmission Electron Microscopy was performed with a TESCAN Mira 3 model STEM mode and back Scattered Electrons in SEM. Fluorescence images were performed with the Olympus BX63 microscope with 365 nm excitation wavelength. Raman measurements were acquired using a HORIBA LabRAM HR Evolution spectrometer with a 633 nm excitation wavelength. X-ray Photoelectron Spectroscopy (XPS) measurements were carried out using a VersaProbe III 5000 photoelectron spectrometer (Physical Electronics), employing Al Kα X-rays with a photon source of 1486.7 eV. Survey scans were collected from 0 to 1400 eV with a pass energy of 226 eV for each sample. Data processing was performed according to Multipack software, applying the Shirley-type background consideration. Curve fitting was performed using a nonlinear algorithm assuming mostly Gaussian peak shape without asymmetries. Survey XPS data were used to examine the atomic composition and surface of the tree species under analysis. After correction with the experimentally determined sensitivity factors, atomic percentage values, and elemental ratios were calculated from the peak-area ratios.

## 5. Conclusions

In summary, we describe the photochemical reduction method to optimize the deposition of silver nanoparticles over the diatom surface, as a potential nano-plasmonic biosensor, providing high-density hot spots with an enhanced optical field for DNA detection. Our findings reveal that excitation radiation at a wavelength of 440 nm promotes the formation of silver nanoparticles on diatoms with uniform size, enhancing the fluorescence properties of the resulting nanocomposite. This discovery has potential applications in developing DNA fluorescence-detecting biosensors, biomarkers, and diagnosis.

## Figures and Tables

**Figure 1 marinedrugs-21-00185-f001:**
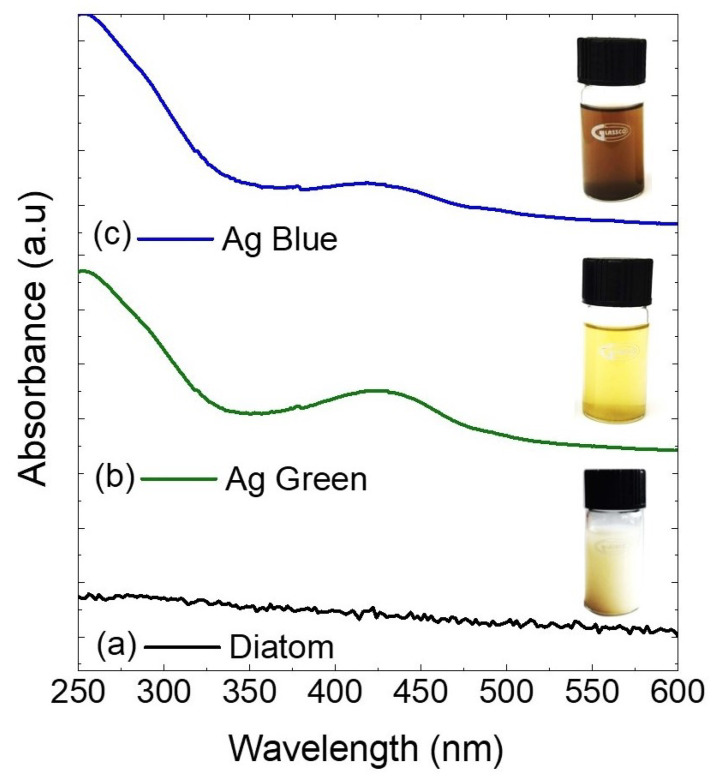
Ultraviolet-visible absorption spectra of (**a**) diatoms irradiated at 540 nm (**b**) and 440 nm (**c**).

**Figure 2 marinedrugs-21-00185-f002:**
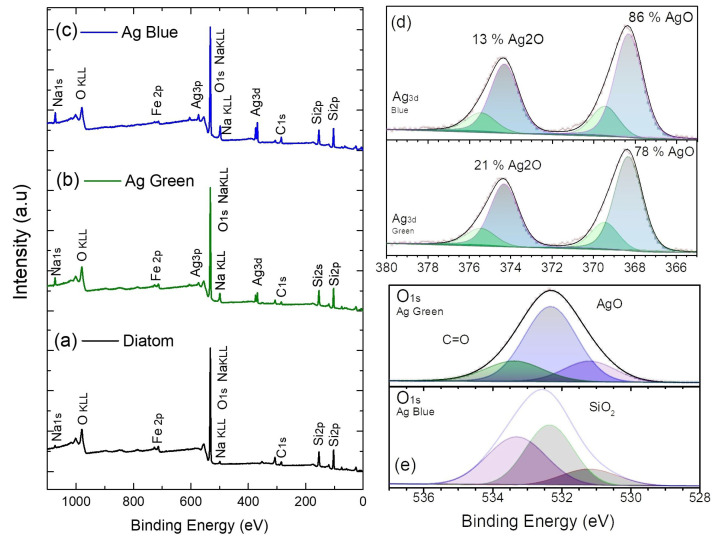
(**a**) X-ray photoelectron spectrum of diatoms; (**a**), diatoms irradiated at 540 nm (**b**) and 440 nm (**c**). High-resolution scan of (**d**) Ag3d and (**e**) O1s.

**Figure 3 marinedrugs-21-00185-f003:**
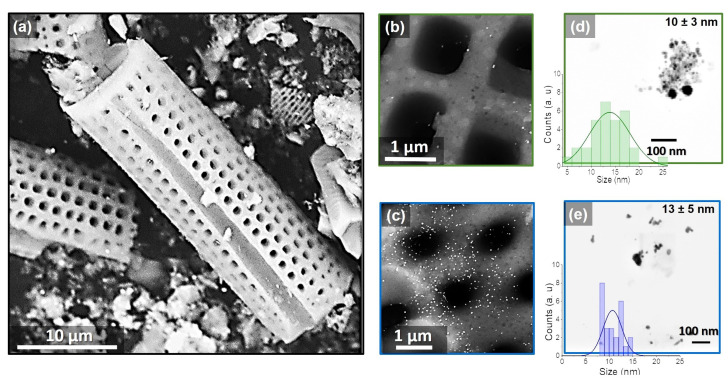
Scanning electron images of (**a**) diatoms, and diatoms irradiated at 540 nm (**b**) and 440 nm (**c**). Transmission electron images and average particle size histograms of diatoms irradiated at 540 nm (**d**) and 440 nm (**e**).

**Figure 4 marinedrugs-21-00185-f004:**
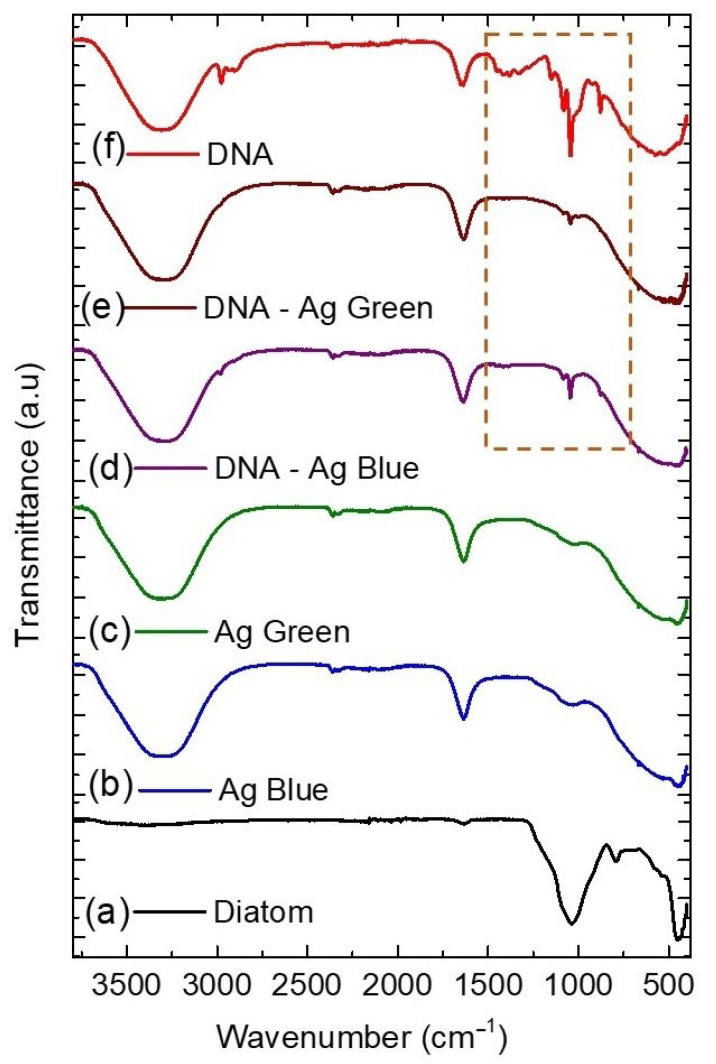
Fourier transform infrared spectra of diatoms (**a**), diatoms irradiated at 440 nm (**b**) and 540 nm (**c**), diatoms at 440 nm (**d**) and 540 nm (**e**) with DNA (dotted square), and DNA pristine (**f**).

**Figure 5 marinedrugs-21-00185-f005:**
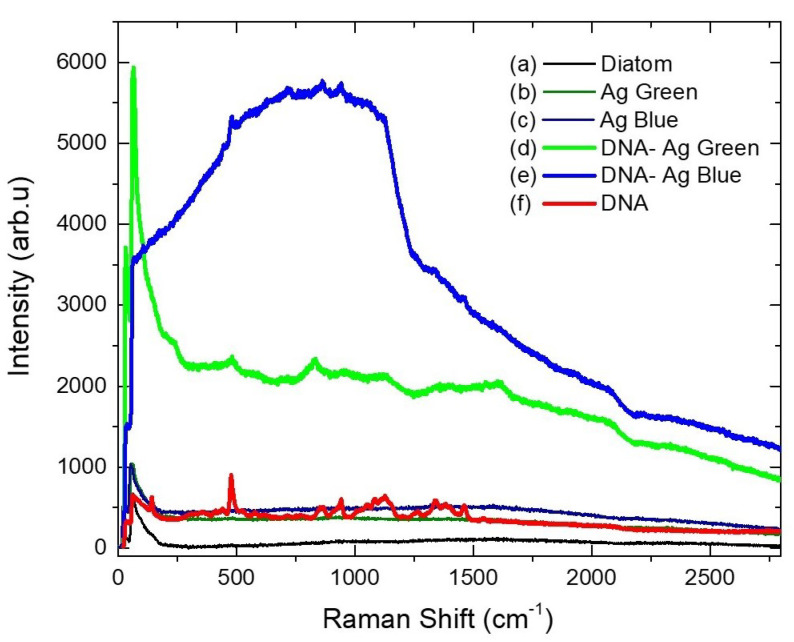
Raman spectra of (a) diatoms, diatoms irradiated at (b) 540 nm (Ag Green), (c) 440 nm (Ag Blue). Diatoms with DNA, irradiated at (d) 540 nm (DNA Ag Green), (e) 440 nm (DNA Ag Blue), and (f) DNA pristine, measured with 633 nm excitation wavelength.

**Figure 6 marinedrugs-21-00185-f006:**
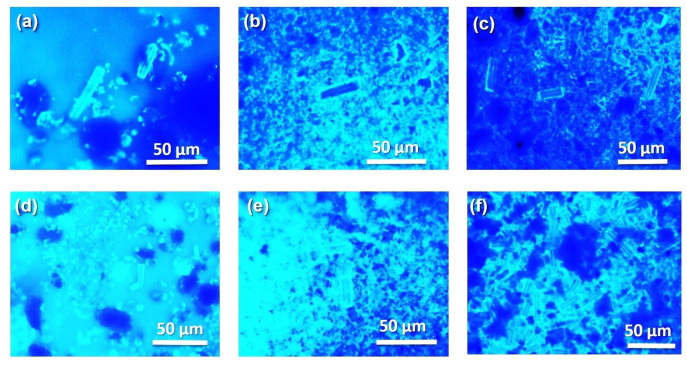
Fluorescence images of (**a**) diatoms, diatoms irradiated at (**b**) 440 nm (Ag Blue), and (**c**) 540 nm, (**d**) diatoms with DNA, irradiated at (**e**) 440 nm (Ag Blue), and (**f**) 540 nm with DNA, measured at 365 nm excitation wavelengths.

**Figure 7 marinedrugs-21-00185-f007:**
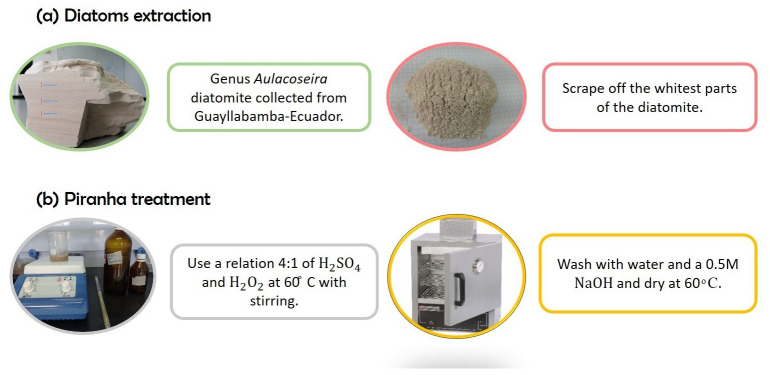
Diatom extraction (**a**) and piranha treatment (**b**).

**Figure 8 marinedrugs-21-00185-f008:**
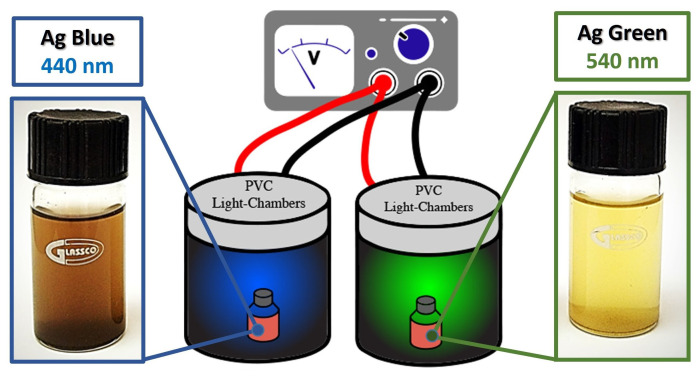
Photoreduction of silver nanoparticles on diatoms.

**Table 1 marinedrugs-21-00185-t001:** Results and assignments of the deconvolution of XPS spectral bands of the diatoms irradiated at 540 nm (Ag Green) and 440 nm (Ag Blue).

Element	Atomic Concentration (%)			Assignments
	Diatom	Ag Green	Ag Blue	
O1s	69.7	68.7	63.7	Si-O, Ag-O, C-O
Si2p	19.1	19.0	18.9	Si-O
C1s	6.5	5.0	11.0	C-O
Ag3d	0	1.1	1.8	Ag-O

**Table 2 marinedrugs-21-00185-t002:** Fourier transform infrared wavenumber of diatoms, diatoms irradiated at 540 nm (Ag Green) and 440 nm (Ag Blue), diatoms with DNA, and DNA pristine.

Diatom	Ag Blue	Ag Green	DNA-Ag Blue	DNA-Ag Green	DNA	Assignment
-	3291	3291	3291	3291	3291	O–H stretching
-	-	-	2974	-	2974	N-H stretch
-	-	-	-	-	2913	C-H stretch
-	1645	1645	1645	1645	1645	O–H/C=O stretch
-	-	-	1044	1044	1044	C-O/C-N
1034	1034	1034	-	-	-	Si-O stretch

**Table 3 marinedrugs-21-00185-t003:** Raman shift of diatoms, diatoms irradiated at 540 nm and 440 nm with DNA, and DNA pristine.

DNA	DNA-Ag Blue	DNA-Ag Green	Diatom	Assignment
480	480	480	-	N-H
950	950	950	-	C-C
1126	1126	1126	-	C-N
1463	1463	1463	-	N-H
1463	1463	1463	-	C-H

## Data Availability

The data supporting this study’s findings are available upon reasonable request from the authors.

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
