# Peer review of "Photochemical Reduction of Silver Nanoparticles on Diatoms"

_marinedrugs, 2023, doi:10.3390/md21030185_

Round 1

Reviewer 1 Report

Dear writers,

Your work seems good, but I have some feddbacks. Firstly in articles generally the contents must be first the introduction. Secondly Materials and methods and then your results and discussion-conclusion. So you have to change the order.

Secondly, as you make a biosensor you have to make a figure about this and in methods you have to give all the parameters that you use in case that someone wants to make around the same experiments to know what to do

Thirdly, I think thata you have to give a better description about your sensor

Last but not least you have to make a table with older works in that sensor and compare with your work, the comparison must be given to your discussion and give in some word in the conclusion the advantages of your work. I am not sure if your methods have Limit of detection, you have to give them and that should be written in a table after your results.

This practice I think that it would give you more references, 24 references are not a lot. So I hope in the second review to see those changes.

Kinde regards

Author Response

Reviewer 1

Dear writers,

Your work seems good, but I have some feddbacks. Firstly in articles generally the contents must be first the introduction. Secondly Materials and methods and then your results and discussion-conclusion. So you have to change the order.

Response: Thanks to the reviewer for your comment. We changed the order of the manuscript following the journal policy,  we received an E-mail from the Editorial Office on 27 Feb 2023 saying:

Before processing your manuscript further, we would be grateful if you could make some changes such that your manuscript follows the Marine Drugs format and style guidelines.

1, Marine Drugs requires Results to be presented before Methods and Materials;

2, The Graphical Abstract should be uploaded as a separate file”.

For that reason, we placed the materials and methods in that order within the manuscript.

Secondly, as you make a biosensor you have to make a figure about this and in methods you have to give all the parameters that you use in case that someone wants to make around the same experiments to know what to do. Thirdly, I think that you have to give a better description about your sensor.

Response: Following the reviewer's suggestion, we change the title of the manuscript from Silver Nanoparticles on Diatoms as DNA Fluorescence Biosensor, to Photochemical Reduction of Silver Nanoparticles on Diatoms. We believe that with this change, we clarify the aim of the article, which is the study of the structural properties of silver nanoparticles on diatoms using the Photochemical reduction method, with some preliminary results related to the potential of these samples to be used as a biosensor.

 Moreover, more results are needed to report the sensor design and the parameters.

Last but not least you have to make a table with older works in that sensor and compare with your work, the comparison must be given to your discussion and give in some word in the conclusion the advantages of your work. I am not sure if your methods have Limit of detection, you have to give them and that should be written in a table after your results.

Response: Following the reviewer's suggestion, we include more references and compare them with our work, as follows:

Our findings agree with previous studies that report the application of the Ag NPs on Pinnularia diatoms for SERS sensing using Rhodmine at 532 nm, obtaining a 4–6× improvement in sensitivity [24]. Then, gold-coated Aulacoseria diatoms were found to be good SERS substrates, enhancing the spontaneous Raman scattering of Mercaptoaniline by a factor of 105 [25]. Finally, there has been reported that natural diatoms biosilica combined with metallic nanoparticles could be used in sensing, diagnostics, and therapeutics, taking advantage of the SERS effect [26].

This practice I think that it would give you more references, 24 references are not a lot. So, I hope in the second review to see those changes.

Response: We include more references in the manuscript as the reviewer suggest.

Reviewer 2 Report

The research work titled ‘Silver Nanoparticles on Diatoms as DNA Fluorescence Biosensor’ by León-Valencia et al., discusses a novel methodology for the green synthesis of AgNPs, where the photoreduction method at 440 or 540 nm excitation wavelengths was used to optimize the growth of silver nanoparticles over the diatom surface. The synthesized nanocomposites were characterized by different spectroscopic and microscopy techniques: UV-vis, Fourier transforms infrared spectroscopy (FTIR), X-ray photoelectron spectroscopy (XPS), Scanning Transmission Electron Microscopy (STEM), Fluorescence microscopy, and Raman spectroscopy to comprehensively understand the process conditions, and the analysis is impressive. The results revealed an enhancement of the fluorescence response 3.5 times higher for the nanocomposite irradiated at 440 nm, than the sample irradiated at 540 nm for DNA detection. Although the results are intriguing, the discussion section is not adequate. This research work is well-planned and thoroughly executed and is suitable for consideration for publication, provided the below mentioned reviewer comments are addressed.

1.     Authors must comment on the lifetime of the fluorescence with and without the presence of nanoparticles. How about the photostability and reproducibility? Please comment.

2.     The figure 1 shows the images of the bottles for AgNPs. The image for diatom also should be shown for comparison.

3.     It is shown that the highest concentration of AgNPs on the diatom surface was obtained for 440 nm sample. The scientific reasoning for such observations should be provided.

4.     The importance of this work is not emphasized with relevant literature. The green synthesis of AgNPs has been demonstrated using biomaterials such as lycoat, gelucire, and soluplus in different publications for "plasmon, fluorescence based biosensing applications". These relevant works should be discussed highlighting on the aspects of go-green approach and green nanotechnology.

5.     Generally, the fluorescence signal overpowers the Raman spectra because of which it is extremely difficult to observe both the enhanced fluorescence and Raman in a single sample. It is impressive that the authors present such results. Authors should clearly explain the experimental parameters adopted to accomplish the same.

6.     How are the enhancements calculated. What samples were used for the blank measurements? Were the polarizers (TM and TE) used to understand the Raman and fluorescence spectra in detail.

7.     It is important to provide the high-resolution TEM images of the AgNPs, displaying the d-spacing/lattice fringes.

8.     The references are not comprehensive. Authors should highlight the synthesis of plasmonic Ag (which resulted in better performance from plasmonic point of view) using light-irradiation using recent references: ACS Appl. Nano Mater. 2023, 6, 2, 1129–1145; Mar. Drugs 2022, 20(1), 56.

Author Response

 Reviewer 2

The research work titled ‘Silver Nanoparticles on Diatoms as DNA Fluorescence Biosensor’ by León-Valencia et al., discusses a novel methodology for the green synthesis of AgNPs, where the photoreduction method at 440 or 540 nm excitation wavelengths was used to optimize the growth of silver nanoparticles over the diatom surface. The synthesized nanocomposites were characterized by different spectroscopic and microscopy techniques: UV-vis, Fourier transforms infrared spectroscopy (FTIR), X-ray photoelectron spectroscopy (XPS), Scanning Transmission Electron Microscopy (STEM), Fluorescence microscopy, and Raman spectroscopy to comprehensively understand the process conditions, and the analysis is impressive. The results revealed an enhancement of the fluorescence response 3.5 times higher for the nanocomposite irradiated at 440 nm, than the sample irradiated at 540 nm for DNA detection. Although the results are intriguing, the discussion section is not adequate. This research work is well-planned and thoroughly executed and is suitable for consideration for publication, provided the below mentioned reviewer comments are addressed.

  1. Authors must comment on the lifetime of the fluorescence with and without the presence of nanoparticles. How about the photostability and reproducibility? Please comment.

Response: Raman spectra were collected in different points on the diatom surface showed a reproducible

scattering pattern and intensity. Preliminary observations shows that the fluorescence of the sample only appears when it is irradiated, and this fluorescence was enhanced with the presence of the silver nanoparticles, presenting a reproducible behavior. We appreciate the reviewer comment to improve the work, and we consider it for future  planning of  experiments to complement the project.

  1. The figure 1 shows the images of the bottles for AgNPs. The image for diatom also should be shown for comparison.

Response: Following the reviewer comment we add the picture of the diatom solution in Figure 1 as follows:

  1. It is shown that the highest concentration of AgNPs on the diatom surface was obtained for 440 nm sample. The scientific reasoning for such observations should be provided.

Response: The growth of the nanoparticles could be explained when the nanoparticles are in resonance with the excitation wavelength forming rapid nucleation of Ag NPs on the diatom surface; this process is known as plasmon-assisted growth [19]. In this process, the λirr dependence control the size and anisotropy of the nanoparticles. As shown in Figure 3(d), larger sizes and a wider distribution are obtained for the samples irradiated at 540 nm wavelength. Meanwhile, the 440 nm excitation wavelength promotes a gradual growth of the nanoparticles along the diatom surface with a uniform narrow distribution [20].

  1. The importance of this work is not emphasized with relevant literature. The green synthesis of AgNPs has been demonstrated using biomaterials such as lycoat, gelucire, and soluplus in different publications for "plasmon, fluorescence based biosensing applications". These relevant works should be discussed highlighting on the aspects of go-green approach and green nanotechnology.

Response: Following the reviewer suggestion we include more references and discussed in the introduction section. 

  1. Generally, the fluorescence signal overpowers the Raman spectra because of which it is extremely difficult to observe both the enhanced fluorescence and Raman in a single sample. It is impressive that the authors present such results. Authors should clearly explain the experimental parameters adopted to accomplish the same.

Response: Raman measurements were acquired using a HORIBA LabRAM 232 HR Evolution spectrometer with a 633 nm excitation wavelength. Data obtained was plotted without any treatment or modification. We agree that most of the times the noise  of fluorescent signals interferes and limits the detection of SERS signals. One of the strategies is changing the excitation wavelength. In this work we did  try different excitation wavelengths, however, we observed  that the fluorescence background did not completely  annihilated the SERS signal. Therefore,  the combination of the diatoms with the plasmonic nanoparticles results in a simultaneous observation of the fluorescence and Surface Enhanced Raman.

  1. How are the enhancements calculated. What samples were used for the blank measurements? Were the polarizers (TM and TE) used to understand the Raman and fluorescence spectra in detail.

Response: The enhancement was calculated by the intensity difference of each spectrum with respect to pure DNA spectrum without nanoparticles. To calibrate the laser, we used a silicon wafer with and without diatoms. We didn’t use polarizers.

  1. It is important to provide the high-resolution TEM images of the AgNPs, displaying the d-spacing/lattice fringes.

Response: We appreciate the reviewer comments to improve the manuscript, but in Ecuador we don’t have access to a high-resolution transmission electron microscopes. In Figure 3 we show the SEM, TEM and histograms of the nanoparticles over the diatoms surface.

  1. The references are not comprehensive. Authors should highlight the synthesis of plasmonic Ag (which resulted in better performance from plasmonic point of view) using light-irradiation using recent references: ACS Appl. Nano Mater. 2023, 6, 2, 1129–1145; Mar. Drugs 2022, 20(1), 56.

Response: We include the recommended references in the manuscript and suggested by the reviewer:

Nd2O3-Ag Nanostructures for Plasmonic Biosensing, Antimicrobial, and Anticancer Applications

Seemesh Bhaskar, Venkatesh Srinivasan, and Sai Sathish Ramamurthy. ACS Applied Nano Materials 2023 6 (2), 1129-1145

DOI: 10.1021/acsanm.2c04643

Younis, N.S.; Mohamed, M.E.; El Semary, N.A. Green Synthesis of Silver Nanoparticles by the Cyanobacteria Synechocystis sp.: Characterization, Antimicrobial and Diabetic Wound-Healing Actions. Mar. Drugs 202220, 56.

https://doi.org/10.3390/md20010056

Round 2

Reviewer 1 Report

Dear writers,

I want to thank you for your comments. I am surprised about the order, but if the editors have this style, it's ok. About the title of the article, I think that is good idea that you change it, as you don't do a research in biosensor, but in properties, but still I think that is good idea to be more decriptive in your methods and you can still put some figure or photos about how you do, for instance the diatom extraction, that you mentioned in materials and methods. It's good that you put more references, but I think that still they are not enough, for a normal article according to my opinion references should be a number between 35-40.

I will not ask another major revision, but is good in order to have a good article with good scientific soudness, you have to be more descriptive around your methods, so if someone doesn;t know for example a lot of things for diatom extraction or photoreduction or other methods you may work, to understand some things. So the extra references can be sth theoritical for your methods and of course you have to describe and put some pictures

Kind regards

Author Response

Ms. Crystal Chen

Section Managing Editor

Marine Drugs

Manuscript ID: marinedrugs-2276246

Dear Editor, thank you for the opportunity to revise the article entitled Photochemical Reduction of Silver Nanoparticles on Diatoms; The suggestions offered by the Editor and the reviewers have been immensely helpful in improving the manuscript. We have included the reviewer's comments and the Manuscript Formatting Request immediately after this letter and responded to them individually, indicating exactly how we addressed each concern or problem and describing the changes we have made. The revised manuscript presents all changes highlighted in yellow. We hope you can consider this revised version for publication in the Biomaterials of Marine Origin section and the special issue: Functional Biomaterials from Marine Diatoms in Marine Drugs.

Thank you for your consideration.

Sincerely,

Dr. Gema Gonzalez

School of Physical Sciences and Nanotechnology

YachayTech University - Ecuador

E-mail: ggonzalez@yachaytech.edu.ec

REVIEWER COMMENTS:

Reviewer 1

Dear writers,

I want to thank you for your comments. I am surprised about the order, but if the editors have this style, it's ok. About the title of the article, I think that is good idea that you change it, as you don't do a research in biosensor, but in properties, but still I think that is good idea to be more decriptive in your methods and you can still put some figure or photos about how you do, for instance the diatom extraction, that you mentioned in materials and methods. It's good that you put more references, but I think that still they are not enough, for a normal article according to my opinion references should be a number between 35-40. I will not ask another major revision, but is good in order to have a good article with good scientific soudness, you have to be more descriptive around your methods, so if someone doesn;t know for example a lot of things for diatom extraction or photoreduction or other methods you may work, to understand some things. So the extra references can be sth theoritical for your methods and of course you have to describe and put some pictures.

Response: We appreciate the reviewer’s comments to improve the manuscripts, and now we include more references, extend the description of the methodology and complement it with figures.

Reviewer 2 Report

The authors have addressed reviewer comments.

Author Response

(The authors gave the same response as above.)
